# Inorganic Nanomaterials in Tissue Engineering

**DOI:** 10.3390/pharmaceutics14061127

**Published:** 2022-05-26

**Authors:** Eleonora Bianchi, Barbara Vigani, César Viseras, Franca Ferrari, Silvia Rossi, Giuseppina Sandri

**Affiliations:** 1Department of Drug Sciences, University of Pavia, Viale Taramelli 12, 27100 Pavia, Italy; eleonora.bianchi04@universitadipavia.it (E.B.); barbara.vigani@unipv.it (B.V.); 2Department of Pharmacy and Pharmaceutical Technology, University of Granada, Campus Universitario de Cartuja, 18071 Granada, Spain; cviseras@ugr.es

**Keywords:** nanomaterials, clays, bioceramics, magnetic nanoparticles, metal oxides, metallic nanoparticles, tissue engineering

## Abstract

In recent decades, the demand for replacement of damaged or broken tissues has increased; this poses the attention on problems related to low donor availability. For this reason, researchers focused their attention on the field of tissue engineering, which allows the development of scaffolds able to mimic the tissues’ extracellular matrix. However, tissue replacement and regeneration are complex since scaffolds need to guarantee an adequate hierarchical structured morphology as well as adequate mechanical, chemical, and physical properties to stand the stresses and enhance the new tissue formation. For this purpose, the use of inorganic materials as fillers for the scaffolds has gained great interest in tissue engineering applications, due to their wide range of physicochemical properties as well as their capability to induce biological responses. However, some issues still need to be faced to improve their efficacy. This review focuses on the description of the most effective inorganic nanomaterials (clays, nano-based nanomaterials, metal oxides, metallic nanoparticles) used in tissue engineering and their properties. Particular attention has been devoted to their combination with scaffolds in a wide range of applications. In particular, skin, orthopaedic, and neural tissue engineering have been considered.

## 1. Introduction

In today’s medicine world, there is an increasing demand for promising biomaterials, which could lead to more accurate treatments. Tissue engineering is gaining great interest as it represents a multidisciplinary approach for creating 3D polymeric substitutes, with the ultimate aim to induce repair and regeneration of injured tissues. However, the polymeric materials alone often result in poor elasticity and resistance to mechanical stress, leading to scaffold failure and/or inadequate cell adhesion and proliferation. For this reason, new nanomaterials need to be explored in order to improve the scaffolds’ mechanical properties and biocompatibility.

Inorganic nanomaterials have recently gained great attention in tissue engineering, in particular to dope polymeric scaffolds. In fact, they have unique properties, such as magnetic and antibacterial ones, and thus they have been widely used to improve tissue conduction, to support and enhance the cell growth, to load drugs, or to guide magnetic and thermal pulses [1]. Inorganic materials can be classified based on their interaction with the tissues as bioinert, bioactive, or bioresorbable materials. Bioinert inorganics are generally used as structural prosthesis and do not interact with the tissues; on the other hand, bioactive inorganics can create bonds with the adjacent tissues, while the bioresorbable materials can be gradually absorbed in vivo and, consequently, they are replaced by the newly formed tissues over time [2].

Inorganics are generally used in tissue engineering as nanomaterials (nanoparticles, smaller than 100 nm in at least one dimension [3]), since materials in nanoscale can efficiently support biological responses. In particular, they can interact with the biomolecules onto the cell surface and be taken up into the cytoplasm. In fact, nanoparticles can affect biological systems in many ways, by stimulating cellular metabolic pathways, as well as by penetrating the cell membrane resulting in the change of cellular activity [1,4].

Due to their biological properties and high surface area to volume ratio, inorganic nanomaterials can be used as drug delivery systems, for antibody labeling, bio-imaging, and tissue regeneration (Figure 1). In particular, the development of scaffolds doped with inorganics allows to obtain suitable mechanical and physiological properties, that could result in the formation of composite structures with adequate strength, osteoconductivity, and bioresorbability [2,5].

The inorganic materials widely used as main factors for tissue regeneration could be grouped in few categories: clays, carbon-based nanomaterials, metal oxides, such as bioceramics, bioglasses, and magnetic nanoparticles, and the metallic nanoparticles.

Carbon-based nanomaterials, metal oxides, and metallic nanoparticles possess high strength and a low elasticity modulus, which make them suitable to repair or substitute damaged hard tissues, such as part of the musculoskeletal system. Moreover, bioceramics are generally considered for their biocompatibility, osteoconductivity, and osteogenic capacity [3,6].

This review will be focused on the description of promising inorganic nanomaterials, in particular clays, carbon-based nanomaterials, metal oxides, and metallic nanoparticles, and their application in different fields of tissue engineering, from soft tissues to hard tissues. The effect of these materials on the scaffolds’ physico-chemical properties has been reviewed; in particular, the enhancement of elasticity and resistance to mechanical stress has been taken into account. Along with these, the safety and the efficacy of these nanomaterials in preclinical models have been considered with a particular attention to the cell behavior and immune response.

## 2. Clays

Natural or synthetized clays have been widely used as pharmaceutical and cosmetic ingredients. In particular, clay minerals are considered useful excipients for the production of drug delivery systems, due to their capacity to interact with drug molecules, their good biocompatibility and swelling properties, low toxicity, and low cost [7]. However, recent studies investigated new and advanced applications of these materials, suggesting the potential of clay minerals to provide new approaches for cell-based regenerative strategies due to their beneficial effects on cellular adhesion, proliferation, and differentiation [8,9].

Phyllosilicates are a group of clay minerals, that can be natural or synthetic, and typically present a layered silicate structure [10].

From a chemical point of view, clay minerals are hydrated aluminosilicate consisting of aluminum and silicon oxides, and they also contain a great number of cations such as Mg^2+^, K^+^, Ca^2+^, Na^+^, and Fe^3+^. From a structural point of view, clay minerals are formed by stratified layers generally constituted by continuous tetrahedral and octahedral sheets. Each tetrahedron is formed by a cation, usually Al^3+^, Si^4+^, and Fe^3+^, and is linked to the adjacent tetrahedra through the basal oxygen atoms to form a two-dimensional hexagonal pattern (Figure 2a). On the other hand, the octahedrons are connected by sharing edges to form hexagonal or pseudohexagonal layers (Figure 2b). These are generally formed by metal cations such as Al^3+^, Fe^3+^, Mg^2+^, and Fe^2+^. The apical oxygen atom of the tetrahedral sheet (T) connects the tetrahedral and octahedral (O) sheets [10,11,12,13].

Depending on the type of association that occurs between the layers, clay minerals can be classified as 1:1, (T:O) or 2:1 (T:O:T) (Figure 2c).

The arrangements of the layers and their physical connections could lead to different morphologies, such as fibers, tubules, laths, and plates.

In particular, it is possible to differentiate between planar clay minerals, such as montmorillonite, hectorite, or laponite, rolled clay minerals, such as halloysite nanotubes, and fibrous clay minerals, such as sepiolite and palygorskite or attapulgite.

The planar clay minerals (Figure 3a) are characterized by interlayer spaces between each T:O or T:O:T stack, which are of great interest as they provide useful properties. Due to this space, the clay minerals are able to adsorb water, leading to an increase in the volume occupied by the clay-water suspension and, consequently, to a swelling in aqueous environments, behaving as hydrogels. Under certain conditions, the layers can also completely delaminate, leading to clay mineral exfoliation [12,13].

On the other hand, the halloysite nanotubes, rolled clay minerals, result from the curvature of newly formed halloysite silicate layers in presence of water, which form the characteristic tubular morphology (Figure 3b).

Lastly, fibrous clays are characterized by a continuous tetrahedral sheet, unlike the octahedral, which is discontinuous (Figure 3c). The apices of the tetrahedral sheet point in different directions, forcing spatial modifications of the discontinuous octahedral sheet, which create channels into the structure. Charges are balanced along the channels by protons, water, and exchangeable cations [13].

The clay minerals most frequently used are natural products derived from mining, which means that they are not subjected to treatments. For example, the phyllosilicates (such as kaolinite, talc, montmorillonite, hectorite, saponite, and sepiolite), and the tectosilicates (such as zeolites) are the most used in pharmaceutical preparations. On the other hand, in some cases clays are subjected to purification or thermal treatments, aimed to improve specific physical or physicochemical properties, purity, or even to change their behavior.

Lastly, synthetic clays can be produced from inorganic salts containing the typical inorganic groups to mimic natural clays [12].

The mechanisms that are involved in the interaction between the natural clays and the organic molecules are various, and they mainly depend on the clay type and on the functional groups of the organic molecules involved. The most common mechanism is the ion exchange caused by charged sites present on clay sheet surface. The exchange is mainly pH-dependent, and it involves anions and rarely cations. Other mechanisms could affect clay minerals interactions, such as drug concentration, pH, temperature, electrolyte concentration, and dielectric constant of the medium [10].

### 2.1. Tissue Engineering Applications

Clay minerals are often associated to organic materials to be employed in the biomedical and tissue regeneration fields. The most useful modification implies the combination between clays and polymers to improve properties of both, such as swelling, rheology, mechanical properties, and cellular uptake [10,13].

Various clays represent promising materials for the application in tissue regeneration, in particular when applied in skin and orthopaedic fields.

#### 2.1.1. Skin Applications

Clay-based scaffolds for skin regeneration represent an interesting option to develop a suitable environment for dermal cell homing [14]. Thanks to their interesting properties, many studies have been focused on clay minerals in skin tissue engineering, and several authors underlined their efficacy. The scaffolds have been manufactured using different techniques such as lyophilization or electrospinning. In particular, Naumenko et al. employed the freeze-drying technology for the fabrication of porous biopolymer systems (based on chitosan agarose and gelatin) combined with halloysite nanotubes. The addition of the clay improved the mechanical properties and wettability of the scaffold. Moreover, the nanocomposite systems showed great biocompatibility in a murine model in vivo, without rejection of the implants and with a complete degradation in six weeks [15].

Yu et al. developed hybrid nanofibers by means of electrospinning, incorporating amoxicillin in a montmorillonite-poly(ester-urethane) urea system. The systems loaded with montmorillonite presented an increase in mechanical properties and, most importantly, they were characterized by a sustained drug release of amoxicillin, which resulted in an antibacterial activity on a murine model in vivo. In fact, these systems were promising candidates in both tissue regeneration and antimicrobial drug release [16].

Moreover, clays and biopolymers have been associated in simpler system obtained by gelation. For this purpose, Massaro et al. designed ciprofloxacin carrier systems based on hectorite/halloysite hydrogels for wound healing applications. Rheological measurements highlighted that the introduction of modified halloysite into the gel matrix improved its properties helping gel formation. Moreover, ciprofloxacin kinetic release tests showed a slow pH-dependent release and MTT test proved the absence of cytotoxic effects on normal human fibroblast cell lines [17].

In a paper of ours, polymer films loaded with a carvacrol (CVR)/clay hybrid (HYBD) were developed for the delivery of CRV in infected skin ulcer treatment. The incorporation of CRV in palygorskite (PHC) reduced its volatility, preserving its antioxidant properties. HYBD showed 20% *w*/*w* CRV loading capacity and was characterized by improved antimicrobial (against S. aureus and E. coli) and cytocompatibility (towards human fibroblasts) properties with respect to pure CRV. Films were prepared by casting an aqueous dispersion containing poly(vinylalcohol) (PVA), poly (vi-nylpyrrolidone) (PVP), chitosan glutamate (CS), sericin, and HYBD. Upon hydration, they formed a viscoelastic gel able to protect the lesion area and to modulate CRV release [18].

#### 2.1.2. Orthopaedic Applications

The inorganic character and the tridimensional organization of clays render them particularly interesting in tissue replacement and regeneration in the orthopaedic field, suitable to guarantee an adequate hierarchical structured morphology and high mechanical properties capable to stand the stresses during new tissue formation [19]. For these reasons, they are promising bioactive materials for mineralized tissue applications [20]. The nano-clay bioactivity is increased, especially when nanocomposites are obtained with superior physical and mechanical properties. Moreover, clays are osteo-inductive in stem cell culture. These findings have been confirmed in a few papers. In particular, Gaharwar A. et al. showed that the PCL (polycaprolactone)/clay-based scaffolds promoted osteogenic differentiation on stem cells (MSCs) by increasing alkaline phosphatase activity along with the production of mineralized matrix. The osteogenic effect increased in a concentration-dependent manner with a maximum at 10% *w*/*w*. Cell proliferation was correlated with the tensile modulus and an appropriate mechanical strength for stem cell growing was recorded between 1–5 MPa [21]. Kundu et al. investigated the fabrication of composite nanoclay-hydroxyapatite-PCL fibers for bone tissue regeneration. The results demonstrated that mesenchymal stem cells (MSCs) were able to thrive and differentiate onto the scaffolds. In fact, they observed calcium deposition, and also collagen formation, which are the main components of the extracellular matrix. The addition of montmorillonite combined with hydroxyapatite resulted in an increase in cell viability and proliferation. Moreover, the osteogenic differentiation of MSCs increased in presence of the scaffold loaded with the clay mineral [22]. Kazemi-Aghdam and co-workers developed an injectable chitosan-modified halloysite nanotubes hydrogel with enhanced mechanical strength and improved osteoinductivity for bone tissue engineering. Interestingly, they encapsulated MSCs into the hydrogels, resulting in enhanced cell proliferation and bone differentiation [23]. Huang et al. also investigated a hydrogel incorporated with halloysite nanotubes fabricated by using the photopolymerization method for potential bone tissue engineering applications. The incorporation of halloysite nanotubes led to an improvement in mechanical properties while maintaining a good cytocompatibility in vitro. Moreover, the halloysite loading upregulated the expression of osteogenic differentiation-related genes and proteins of human dental pulp stem cells, therefore facilitating bone regeneration in calvarial defects of rats [24].

Clays are promising bioactive materials not only for hard but also for soft tissue regeneration [20]. In fact, Bonifacio et al. proved the positive impact of clay minerals incorporation into the scaffolds also for cartilage repair. In particular, mesoporous silica provided the best combination in terms of mechanical properties, morphology, and in vitro cytocompatibility. Moreover, the clay-loaded scaffolds, based on gellan gum, successfully supported the chondrogenic differentiation in a 3D culture model. These systems were also able to enhance the antibacterial response in a murine model in vivo [25].

## 3. Carbon-Based Nanomaterials

Carbon-based nanomaterials are nano composites with a high surface area to volume ratio and a small size, generally between 1 and 100 nm. They have recently gained attention due to their unique characteristics, such as chemical stability, low friction coefficient, thermal and wear resistance, high conductivity, and hardness. Moreover, these can be functionalized at a mass production level rendering them suitable fillers for tissue engineered scaffolds [26,27,28].

These characteristics have made them of great interest in different fields such as imaging, sensing, regenerative medicine, and drug delivery. In particular, carbon-based nanomaterials have been demonstrated to enhance bone regeneration and mechanical properties [26,28,29].

For this purpose, carbon nanotubes (CNTs) and carbon nano-onions (CNOs) have been here described, as they recently became an interesting alternative to reinforce the mechanical, thermal, and antimicrobial properties of various polymers.

### 3.1. Carbon Nanotubes (CNTs)

Carbon nanotubes (CNTs) have emerged from the variety of carbon-based nanomaterials as interesting candidates for the enhancement of the tissue-engineered constructs’ mechanical and biological properties.

CNTs are nanocylinders made of carbon, which can be produced by means of several routes, such as chemical vapor deposition, laser ablation and arc discharge [30]. They can be composed by different numbers of walls, which can be single, with diameters between 0.7 and 2 nm, double, or multiwalled, with diameters up to 100 nm (Figure 4). This wide variability can provide them with different properties interesting for biological, technological, and material applications, such as drug delivery, ultrasound imaging, and biosensors [28,30]. In particular, CNTs possess unique properties useful for the improvement of the physical and the biological performance of the scaffolds for tissue regeneration, such as high thermal (5 × 103 W/m/K) and electrical conductivities (>100 S/cm^2^). Moreover, they also possess great tensile strength, up to 200 GPa, and elastic modulus up to 1.34 TPa, fundamental for the enhancement of the mechanical properties for orthopaedic scaffolds. In fact, CNTs have been used to improve the physical and the mechanical properties of polymers such as polystyrene (PS), poly-L-lactide (PLLA), and polycaprolactone (PCL), actually investigated for the production of scaffolds for hard tissues regeneration [31,32,33].

Furthermore, the use of CNTs as nanofillers for tissue-engineered constructs has been widely explored for the enhancement and modulation of the biological response, as they are able to improve the osteoblastic differentiation and new bone tissue formation. In fact, despite the concerns about their biocompatibility, substrates containing CNTs have been demonstrated to support the adhesion and proliferation of various types of cells, such as osteoblasts, neurons, and smooth muscle, also promoting angiogenesis [28,34,35,36].

### 3.2. Carbon Nano-Onions (CNOs)

Carbon nano-onions (CNOs) are a versatile class of carbon-based nanostructures composed of multiple concentric shells of fullerenes, that recently aroused great interest in the scientific community. They are characterized by a cage-within-cage structure, with smaller fullerenes inside larger ones (Figure 5) [26,38]. They consist of quasi-spherical nested graphitic layers close to one another with dimensions that can range from 2 to 50 nm, depending on the synthesis method. The distance between the shells is 3.4 Å, which is slightly different to the distance between two graphitic planes, that is 3.334 Å [39]. The pentagonal and hexagonal rings that compose the structure consist of two single bonds and one double bond between the carbon atoms. Moreover, they can contain either C_60,_ a hollow core, or a metallic core as the innermost shell [26,39,40].

Interestingly, the synthesis method can determine CNOs physico-chemical properties that are related to their shape, dimensions, the number of layers, and the distance between them. In fact, CNOs can be categorized based on their size, shape, and type of core as:(a)small-sized, which are below 10 nm, or big-sized, which are above 10 nm;(b)spherical or polygonal;(c)dense, which are characterized by a core filled with different metals, or hollow, which have an empty core.

CNOs unique electronic and structural properties make them great candidates for various applications. In fact, rapid development of these structures has recently occurred due to their remarkable physico-chemical properties, such as large surface area, high thermal stability, broad absorption spectra, and ability to reversibly accept multiple electrons [38]. Their surface can also be modified with fluorophores or other ligands for imaging and targeted drug delivery applications.

Moreover, they have been demonstrated to possess exceptional biocompatibility and biosafety in vivo, representing an attractive choice as nanofillers for tissue-engineered scaffolds and biological systems. CNOs have been demonstrated to possess low toxicity, high pharmaceutical efficiency, and high dispersity degree in aqueous solutions, due to the introduction of surface functional groups. Furthermore, in preclinical models, they show the capability to remain in systemic circulation for hours with weak inflammatory potential, making them exceptional drug carrier candidates [26,41,42].

### 3.3. Tissue Engineering Applications

As mentioned, CNTs and CNOs have gained great interest in the scientific community for their ability to increase the scaffolds mechanical properties (Table 1) and biocompatibility in the field of the bone tissue regeneration. However, since there are a few findings of in vitro and in vivo safety in the biological conditions, the biocompatibility is a key issue that deserves to be explored. Despite this, there is some evidence in the literature concerning the development of scaffolds for tissue reparation and some of those report the in vivo results obtained in preclinical model.

#### 3.3.1. Skin Applications

Generally, the scaffold doping using carbon-based nanomaterials has the overall aim to reinforce the polymer based structure, increasing elastic behavior. A matrix based on chitosan and polyvinyl alcohol (PVA) loaded with oxidized CNOs (CS/PVA/ox-CNO) was developed by Tovar et al. as a nanocomposite film for tissue engineering application. They demonstrated that the introduction of ox-CNOs enhanced the stability of the CS/PVA scaffold. The mechanical properties of the matrix enriched with CNOs significantly improved. In particular, an increase in the ox-CNO content led to an increase of about 38% in Young’s modulus and 27% in tensile strength. Moreover, the biocompatibility was tested in vivo using rat subcutaneous tissue implantation. No allergic response or pus formation was observed in the rats after 30 days of implantation. In all cases, repair of the surgical defects and hair were observed, demonstrating a normal biosorption process. The histological study also demonstrated that the scaffold was biocompatible and biodegradable, even with a high content of ox-CNO. Hence, the CS/PVA/ox-CNO scaffold demonstrated tissue regeneration capability [41].

Patra et al. developed polyaniline (PANI) nanofibers enriched with CNTs to regenerate tissues providing the hydrophobicity/hydrophilicity balance that could supply nutrients and growth factors to the seeded cells. The scaffold showed excellent biocompatibility in vitro on fibroblasts. Moreover, the system demonstrated sensitivity to inflammation and capability to respond to loco-regional acidosis that delay the wound healing process, representing a suitable option for cell grafting and tissue regeneration [47].

#### 3.3.2. Orthopaedic Applications

The carbon-based nanomaterial doping of scaffolds for the orthopedic tissue not only increases the elasticity of the systems but also the hardness and the resistance to compressive forces. Pan et al. developed a composite scaffold based on polycaprolactone (PCL) and reinforced with multi-walled CNTs. The addition of the CNTs increased the tensile strength and compressive moduli of the scaffolds. The compressive modulus increased by 54% compared to the systems of PCL alone, while the tensile modulus increased from 85 to 100 MPa. Moreover, the scaffolds could promote the proliferation and differentiation of rat bone-marrow-derived stroma cells more than pure PCL control group, representing potential tools to be used in bone tissue regeneration [32]. Collagen sponges with a multi-walled CNTs coating have been also developed for bone tissue engineering by Hirata et al. Primary rat osteoblasts were cultured onto the matrices and calcium and osteopontin contents were evaluated. The osteoblasts grown on the CNTs-coated sponge differentiated earlier in respect to the uncoated one, demonstrating favorable biocompatibility with bone. Moreover, the calcium and osteopontin content of the CNTs-coated matrix after seven days was significantly higher than that of the uncoated one. Therefore, they concluded that CNTs coating of a 3D collagen scaffold would be suitable for bone tissue engineering [48]. In a further study, Duan et al. incorporated CNTs in poly(L-lactide) acid (PLLA) nanofibrous scaffolds to investigate its osteoinductive properties. The scaffold enriched with CNTs demonstrated remarkable cell adhesion and proliferation, and significantly increased the osteogenic differentiation of bone mesenchymal stem cells in respect to the not-loaded scaffold. Moreover, in vivo experiments also revealed that CNTs-loaded matrices could induce osteogenesis and remarkably enhance the expression of both osteogenesis-related proteins and type I collagen more than pristine PLLA matrices [49].

#### 3.3.3. Neural Applications

Also in neural application, the scaffold doping with carbon-based nanomaterials has the primary goal to reinforce the structure. Fundamental for this application is the carbon-based nanomaterials conductivity that act as a template to guide cell proliferation. PLLA-CNT based scaffolds have been investigated to modulate neuronal differentiation by Scapin et al. The systems demonstrated good mechanical flexibility, necessary for implant purposes, full biocompatibility, and support of cell growth. In particular, the scaffolds supported cell adhesion and neuronal differentiation better than pristine PLLA ones, representing interesting candidates for implantable systems for autologous neuronal differentiation [50]. Gupta et al. developed aligned chitosan scaffolds combined with multi-walled CNTs by means of electric field alignment technique. The aligned conformation improved the elastic modulus, yield strength, and ultimate tensile strength by 12.7%, 21.9%, and 11.2%, respectively, if compared with the random structure. Moreover, the alignment of the CNTs led to higher anisotropic electrical conductivity along fibers direction, which is fundamental for the cell guidance in the right direction. The CNTs-loaded scaffolds were biocompatible with an increase in viability. Moreover, 50-60% of neurons were found to be aligned in the CNTs alignment direction of the matrices, which could result in a repopulation of regions characterized by acute neuronal loss [51].

CNTs were also studied dispersed within collagen hydrogels by Lee et al. to provide suitable microenvironmental conditions for stimulating mesenchymal stem cells in neural regeneration. The CNTs not only did not induce toxicity to the mesenchymal stem cells, but also improved their proliferative potential. The CNTs-loaded hydrogels also increased the cells expression of neural markers, and significantly promoted neurotrophic factors, in particular nerve growth factor and brain derived neurotrophic factor, leading to enhancement in neurite outgrowth behaviors [52].

## 4. Metal Oxides

Metal oxides have been studied over the years for different applications, such as catalytic, dielectric, electromechanical, and only recently research studies have been targets to explore their use in biomedical applications [53,54].

Metal oxides can exist in different shapes and sizes, which are correlated to their synthesis. When they present nano-dimensions, ranging from 1 to 100 nm, they are characterized by unique properties, particularly interesting for biomedical application. Nanoparticles having diameters less than 20 nm could more easily enter the cell membrane and the cellular organelles and could also cross the brain barrier. Moreover, they could even penetrate the bacterial cells and release toxic metal ions [55].

Ideally, the metal oxide nanoparticles for biomedical applications should be chemically stable, should not easily dissociate in metal ions, should not present toxicity, correlated to the size and surface properties, and lastly and importantly, they should be biocompatible [56,57]. To assure these characteristics, the use of metal oxides has recently reached internationally recognized standards. For example, zirconium dioxide must be prepared following international standard reference ISO 13356, which specifies the requirements and the related test methods for the production of biocompatible nanomaterials intended for biomedical applications [54]. The most important examples of metal oxide nanoparticles employed in biomedicine are represented by bioceramics and bioglasses and by magnetic nanoparticles.

### 4.1. Bioceramics and Bioglasses

The bioceramics are biomaterials used to treat, augment, or replace the damaged tissues, in particular the hard ones. They are characterized by properties that make them body friendly substitutes, such as biocompatibility, degradation, and high mechanical strength, suitable to improve the mechanical properties of the scaffolds.

The success of bioceramics in biomedicine is due to their biofunctionality and biocompatibility. In fact, the formation of apatite on their surface after implantation makes easier the bonding of the substrates with the body tissues [58].

The bioceramics are mainly classified according to the tissue response into three subclasses: bioinert, bioactive, and bioresorbable ceramics (Figure 6) [59,60,61].

Bioinert ceramics are characterized by stable physicochemical properties and good biocompatibility especially with the hard tissues. They keep both their mechanical and physicochemical properties once implanted in the host, without causing an immunological rejection. The main use of bioinert materials is the production of structural supports, such as bone plates or screws, due to their ability to resist fractures [58]. In particular, alumina and zirconia, both alone and combined, are traditionally used for dental and orthopaedic applications, due to their improved mechanical and morphological properties and suitable biocompatibility [62,63].

Bioactive ceramics, such as bioglasses or glass ceramics, can interact with the host tissues, inducing a specific biological response that improves tissue regeneration [58,64]. Once implanted, they form a hydroxyapatite layer similar to the inorganic phase of the native bones, which can bond both the collagen fibrils of the tendons and the bone. Glass ceramics are mainly constituted by CaO and P_2_O_5_, which are also the main constituents of the bone mineral phase. For this reason, they are characterized by an optimal effectiveness. Bioglasses are widely used as implants, as they have a positive effect on living cells and tissues, due to chemically stable bonds with the skeletal system of the host. Moreover, the microstructure of bioglasses increases the bending strength and the compressive strength of the implanted material, and they also have been referred as enhancers of angiogenesis, which is a crucial step for the wound healing process [65,66,67,68].

Lastly, bioresorbable ceramics interact with the host tissues and are also able to degrade rapidly once they meet the biological fluids. Their chemical structure is broken by the tissue fluids, and they are completely absorbed by the body without producing any toxic effect. For this reason, they do not require second surgery for implant removal. The main materials of this class are hydroxyapatite (Ca_10_(PO_4_)_6_(OH)_2_) and calcium phosphates, which have been used mainly in orthopedics as bone substitutes, due to their stability, biocompatibility, and osteo-conductivity [58,68,69].

### 4.2. Magnetic Nanoparticles

The magnetic nanoparticles recognized as non-toxic in the medical field, thanks to their oxidative stability, are mainly represented by magnetite (Fe_3_O_4_) and maghemite (Fe_2_O_3_).

Magnetic nanoparticles have been extensively used for biomedical applications with different purposes, such as magnetic resonance imaging (MRI) diagnosis, drug delivery control, cell/tissue targeting, and hyperthermia in cancer treatment [68,70,71]. Recently, there is evidence that their inclusion into scaffolds results in unique properties to control cell signaling both in vitro and in vivo, in particular when they have small size (<100 nm) and narrow size distribution [72].

Interestingly, iron oxide nanoparticles are characterized by a superparamagnetic behavior, namely they show magnetism if an external magnetic field is applied. This is of interest as the nanoparticles lose magnetism after removing the field. Moreover, magnetism retention is strongly related to particles size: 10–50 nm nanoparticles can be affected by an external magnetic field [68].

The supposed mechanism of magnetic nanoparticles embedded in scaffolds for orthopaedic reparation is mechano-stimulation. The nano-movement induced by the magnetic field on the scaffolds seems able to cause forces in the range of pN, and cells act in response to those mechanical stimuli according to four major biochemical pathways: ion channels activation, ATP release, contraction of the cytoplasmatic actin and alteration of protein expression in particular FAK (focal adhesion kinase), which is the basis of biochemical signals, that stimulate the cells remodeling and differentiation. In this context, the magnetic scaffolds could guide the mechano-transduction signals allowing deeper tissue reparation [73,74].

### 4.3. Tissue Engineering Applications

Metal oxide nanoparticles has gained attention in the recent decades since they ideally could combine reparative effectiveness with antimicrobial ones [75].

#### 4.3.1. Skin Applications

Bioactive glasses proved to be beneficial for wound healing in skin tissue engineering, due to their capability to stimulate hemostasis, angiogenesis, and fibroblasts proliferation both in vitro and in vivo [67,76]. Different types of formulation and particles have been considered. Wang and colleagues studied bioactive glasses nanoparticles mixed with gelatin for the production of hydrogels intended for wound dressing. The mixture was able to promote faster tissue regeneration and a more effective wound healing, due to the synergic effect of the bioactive glass and gelatin, within seven days after implantation in a murine model [77]. Samadian et al. developed an electrospun cellulose scaffold loaded with hydroxyapatite (HP) by means of electrospinning technique. The results showed that the concentration of HP affected the porosity, water contact angle, water uptake, water vapor transmission rate, and cells proliferation. In vivo studies showed that all dressings had higher wound closure percentage than the sterile gauze, as the control, reaching values of closure of 93.5% [78]. Babitha et al. investigated the stability of TiO_2_ nanoparticles incorporated in a zein-polydopamine nanofibrous scaffold as potential wound dressing material. The scaffold mimicked the network of the natural extracellular matrix (ECM), promoting cells adhesion and proliferation. Moreover, the in vivo evaluation of the wound healing potential proved the system as suitable for wound healing in tissue engineering applications, since complete re-epithelialization was achieved on day 15 in the group treated with the system loaded with TiO_2_ [79].

#### 4.3.2. Orthopaedic Applications

Metal oxides have been widely considered for their capability to stimulate bone tissue repair, due to their bonding to the living tissues once implanted [68,80] and there are in vivo proofs of concept that suggest the rational for the use of metal oxides in orthopaedic tissue engineering.

In particular, Covarrubias and colleagues reported that the incorporation of bioglasses into a chitosan-gelatin matrix showed excellent cytocompatibility and enhanced the crystallization of bone-like apatite in vitro. Moreover, in vivo implantation of the scaffolds into bone defect models demonstrated that the systems significantly increased the amount of new bone production [81]. Li et al. produced a multifunctional poly(citrate-siloxane) (PCS) elastomer loaded with bioactive glass with intrinsic biomineralization activity and photoluminescent properties for potential bone tissue regeneration. The nanocomposite showed significantly enhanced mechanical properties, hydrophilicity, photoluminescence properties, biomineralization activity, improved osteogenic differentiation ability, osteoblasts biocompatibility, and low inflammatory response in vivo [82].

The efficacy of HP was also widely proven in tissue engineering application for tendon and bone regeneration. The combination of HP with various carriers, such as porous [83] and electrospun [84,85] scaffolds or hydrogels [86] showed enhancement of cellular activity. In particular, the research underlined the HP capability of increasing the mechanical properties of the scaffolds and of supporting cell adhesion and differentiation into osteo-like cells, able to produce ECM.

#### 4.3.3. Neural Applications

One of the most promising applications of metal oxides is in nerve tissue engineering and neuroregeneration [68]. Various studies have been performed particularly on magnetic nanoparticles, in order to make cells magnetically sensitive and allow cell migration, proliferation and differentiation (Figure 7) [87,88,89].

Recently, superparamagnetic iron oxide combined with gold nanoparticles were functionalized with nerve growth factor (NGF) for neuron growth and differentiation. The functionalized systems provided higher PC12 neuronal growth and orientation under dynamic magnetic fields compared to static magnetic fields, confirming the potential of non-invasive magnetic neuron stimulation for promoting neuronal growth [90].

Magnetic nanoparticles have also been used by Chang et al. to control collagen fiber orientation in situ by applying an external magnetic field. In vitro, the magnetically activated neurons extended their neurites along the aligned nanofibers. This increased the cell density, and consequently also the NGF concentration, together with myelination. In a rabbit sciatic nerve model, the scaffold showed superior nerve recovery and less muscle atrophy in comparison with autograft [91].

In another study conducted by Vinzant et al., Fe_2_O_3_ was conjugated with a peptide antisauvagine-30 (ASV-30), since iron oxide nanoparticles are capable to efficiently cross the blood-brain barrier. The infusion of ASV-30 reduced anxiety-like behavior of rats through binding to corticotropin releasing factor type 2 receptors. In vivo results demonstrated that systemic application of iron oxide combined with ASV-30 decreased anxiety with no impact on locomotion, representing a novel approach for the peptide delivery across the blood-brain barrier [92].

One example of metal oxide recently recognized for its relevance in nerve reparation is represented by cerium nanoparticles [93]. In fact, it presents both antioxidant and anti-inflammatory effects, which could lead to cell protection and differentiation, when combined with a polymeric scaffold. Marino and coworkers developed gelatin-based nanofibers loaded with cerium oxide nanoparticles. Both topographical and antioxidant cues were confirmed. The feature of the fibers, such as their porosity, high surface area and biodegradability, permitted the ion exchange necessary for the reduction reaction of Ce^3+^ to Ce^4+^ and consequently the antioxidant effect. Meanwhile, the aligned fibers supported the axonal guidance and outgrowth of neuronal cells. They observed that the presence of metal phase at low concentration did not disturb the fiber alignment and size but increased their mechanical properties. These phenomena are fundamental for mediating the mechano-transduction pathways, such as phosphorylation of FAK, cytoskeletal rearrangements, and nuclear deformations, all outcomes observed in the differentiated SH-SY5Y nerve cells grown onto the fibers surface. Moreover, cerium nanoparticles showed beneficial effect not only on ROS control, but also on neural differentiation by releasing β3-tubulin protein [94].

## 5. Metallic Nanoparticles

Metallic nanoparticles represent an attractive tool for various applications in the field of nanotechnology, as they provide a link between bulk materials and molecular or atomic structures [95]. In fact, they are versatile agents currently employed in diagnostics, cancer targeting, tissue engineering, disease treatment, and many more applications, due to their unique physicochemical characteristics: high surface area-to-volume ratio, presence of edges and corners, electron storage capacity, high surface energy, high dangling bonds, and high energy atoms located on their surface area [96,97,98,99,100]. However, it is also of fundamental importance to identify the risks associated with these nanoparticles before their use to minimize the toxicity and maximize their potentialities. In fact, the cytotoxicity of metallic nanoparticles depends on their concentration, on the exposure time and cell sensitivity, and it occurs by the induction of oxidative stress through disturbance of ionic and electronic flux, disruption of the permeability transition pores, and reduction of the level of cellular glutathione [101,102,103,104]. For this reason, it is of fundamental importance the manufacture and the modification of the metallic nanoparticles utilizing different functional groups, which provide conjugation of antibodies, ligands, and drugs [105,106]. Recent research also attempted to use stabilizing agents, such as polyvinylpyrrolidone, polyvinyl alcohol, and polyacrylic acid, which could be adsorbed onto the nanoparticles surface to form a layer that minimizes particle aggregation and enables synthesis of a stable solution of the metallic nanoparticles [107].

The preparation of engineered nanoparticles is of scientific interest to modulate cellular events for tissue engineering applications. For example, metallic nanoparticles uptake is inversely correlated to their particle size. In fact, 30–50 nm particles show higher cellular internalization compared to 50–200 nm particles, influencing the biological function, such as stem cells differentiation and toxicity [107,108]. The surface charge is also implicated with cellular internalization. In particular, a positive charge, unlike a neutral or negative one, can allow a quicker entrance into the nucleus avoiding lysosome degradation [109].

So far, a significant interest for the applications in tissue engineering and regenerative medicine has been directed primarily to gold (Au) and silver (Ag) nanoparticles, also due to their capability of modulation of stem cell proliferation and differentiation.

### 5.1. Au Nanoparticles (Au NPs)

Au NPs are currently widely used in various biomedical applications, such as genomics, clinical chemistry, laser phototherapy of cancer cells, targeted delivery of drugs, and many more (Figure 8), due to the favorable chemical and physical properties, in particular the high stability and facile synthetic preparation techniques [110,111].

Au NPs possess a wide range of unique properties, in particular the tunable optical resonances, easy surface functionalization, and well-controlled size and shape, that make them versatile platforms for different applications [111,112,113,114].

Due to their characteristics, biocompatibility, and low toxicity, Au NPs represent useful systems for tuning stem cell fate, and consequently tissue regeneration. In fact, they proved able to promote the differentiation of mouse embryonic stem cells (ESCs) into dopaminergic neurons, due to the activation induction of the mTOR/p70S6K signaling pathway [115]. Moreover, nanofibrous scaffolds loaded with Au NPs demonstrated to increase the neurite length and axon elongation [116].

The shape, size, and surface characteristics of Au NPs could also induce the osteogenic differentiation of mesenchymal stem cells (MSCs) and human adipose stem cells (hASCs). In particular, 50-70 nm Au NPs promoted the osteogenic differentiation of MSCs and hASCs [107]. In another study, miR-29b-delivered polyethyleneimine (PEI)-capped AuNPs efficiently promoted the osteogenic differentiation of human bone marrow-derived MSCs and MC3T3-E1 cells with almost no toxicity. This ability to induce osteogenic differentiation was evidenced through the upregulation of the genes related to osteogenic differentiation, i.e., alkaline phosphatase, osteopontin, osteocalcin, and Runt-related transcription factor 2 [117].

### 5.2. Ag Nanoparticles (Ag NPs)

Ag NPs are one of the most widely used metallic nanoparticles for biomedical applications due mainly to their antimicrobial properties. In fact, the use of Ag nanoparticles could be a safety measure to prevent bacterial infections, which are a significant risk in tissue engineering [68,118].

There are various mechanisms that the Ag NPs employ to cause their antimicrobial effect (Figure 9). First of all, the Ag can anchor the bacterial cell all and penetrate it, causing structural changes in the cell membrane and causing its death. This occurs by the accumulation of nanoparticles on the cell surface and the consequent formation of cavities [119]. Another mechanism could be the formation of free radicals. Various studies on electron spin resonance spectroscopy evidenced that Ag NPs formed free radicals when in contact with the bacterial cell wall. The free radicals make the cell membrane porous and damage it, leading to the cell death [120,121].

Ag NPs could also modulate the signal transduction in bacteria. In fact, they alter the phosphotyrosine profile of bacterial peptides, which is noted only in the tyrosine residues of the Gram-negative bacteria, leading to the signal transduction inhibition and consequent inhibition of growth [122]. Since Ag is a soft acid, it could also interact with sulfur and phosphorous residues, which are soft bases and are the main components of cell DNA. The effect of nanoparticles on these soft bases leads to DNA destruction and cell death, due to the inhibition of DNA replication [123].

Finally, it has also been suggested that nanoparticles in contact with bacterial cells could release Ag ions, which interact and inhibit several cell functions that damage the cells. In particular, the inhibition of a respiratory enzyme, which occurs by the inactivation of the thiol groups, generate reactive oxygen species that attack the cell [124,125,126].

### 5.3. Tissue Engineering Applications

Au and Ag NPs seem able to effectively guide cell behavior, enhancing cell differentiation and intracellular delivery, and to impart unique properties to scaffolds where they are embedded [127].

#### 5.3.1. Skin Applications

The enhancement of the mechanical properties and the antimicrobial activity towards bacteria and fungi are the main properties of Au and Ag NPs, which make these nanoparticles excellent candidates for the regeneration of chronic wounds highly prone to infections. In fact, metallic nanoparticles could directly act or be functionalized with antibiotics, antioxidants, and ROS scavengers, and directly applied topically in tissues, leading to the improvement of wound healing process [127,128].

Zi-Wei et al. conducted a study to clarify the specific biological mechanisms promoted by Ag NPs loaded in chitosan oligosaccharide/poly(vinyl alcohol) nanofibers for wound healing (PVA/COS-Ag). The human skin fibroblast response was studied treating the cells with the supernatant derived by the nanofiber at different concentrations. They proved that PVA/COS-Ag nanofibers promoted the secretion of fibroblast transforming growth factor TGF-β1, thus improving their adhesion and proliferation by inducing the S and G2/M cycles in cells. Moreover, the nanofiber treatment significantly up-regulated the collagen and fibronectin synthesis in a dose-dependent manner. Finally, the authors confirmed that PVA/COS-Ag nanofibers activated the key signal TGF-β1/Smad transduction, that is an important pathway affecting the early stages of wound healing [129].

Tian et al. studied in vivo the impact of Ag NPs on burn and diabetic wounds as potential wound healing enhancer. They found that the delivery of Ag both showed antimicrobial effect and increased healing rate. They provoked deep burns, normally cured after 35.4 ± 1.29 days, in male BALB/c mice, then treated a group with silver sulfadiazine (SSD) and the other with Ag NPs (ND). The treatment with SSD determined a slowing down of the healing period to 37.4 ± 3.43, while Ag NPs enhanced the healing process to 26.5 ± 0.93 days (Figure 10).

They also discovered that Ag NPs could regulate the cytokines associated in burn wound healing. Significant decrease in neutrophils was found in wounds treated with ND compared to SSD groups, which indicate effect of Ag NPs to decrease the local and systemic inflammatory response [130].

Wang et al. studied the combination of Ag NPs with poly(gamma-glutamic acid) (g-PGA) hydrogel copolymer to improve the development of wound dressings. They found that hydrogels could continuously release antibacterial factors. Moreover, the system demonstrated to promote wound healing in vivo on male BALB/c mouse in comparison with control groups. Histological analysis revealed collagen deposition and intact epidermis layer formation were observed after 14 days of impaired wound healing [131].

#### 5.3.2. Orthopaedic Applications

Particularly interesting in orthopedics is Au NPs, recently proposed in tissue engineering to enhance bone regeneration, due to their potential to promote cell differentiation [132].

In this frame, Heo et al. developed a biodegradable hydrogel (based on methacrylated gelatin) loaded with Au NPs for bone tissue regeneration. The system promoted in vitro the proliferation, differentiation, and alkaline phosphate activities of human adipose-derived stem cells, which differentiated in osteoblasts cells. Moreover, the hydrogels loaded with high concentrations of Au showed in vivo osteogenic differentiation and consequent new bone formation [133]. Analogously, Li et al. conjugated 2,2,6,6-Tetramethylpiperidine-*N*-oxyl (TEMPO) with 40 nm Au NPs. The conjugated was used to investigate the effect on ROS scavenging, proliferation, and differentiation of MSCs. The systems were efficiently taken up by MSCs and reduced the overproduction of ROS at low dosages. Moreover, they enhanced osteogenic differentiation of MSCs while inhibiting the adipogenic differentiation. Consequently, it could be used for ROS-induced dysfunctions while regulating the desired differentiation type [134]. Similarly, del Mar et al. reported the impact of Au NPs on MSCs migration and proliferation. MSCs were able to colonize fibrin and PCL-based scaffold and osteogenic differentiation was observed in comparison with the untreated cells used as a control [135].

#### 5.3.3. Neural Applications

Au NPs have been also investigated in neural application since they are characterized by a surface that could be easily conjugated with biomolecules facilitating the targeted delivery of growth factors, such as nerve growth factor (NGF) and genes (DNA and RNA) [136]. In this context, Au NPs immobilized with silica spheres were investigated by Park et al. to deliver electrical stimulation to nerve cell cultures in vitro. If the silica supports were of about 100 µm diameter, the PC 12 cells extended neurites on the Au NPs in presence of an electrical stimulation, whereas if the silica supports were lower in dimensions (20 µm) the neurite outgrowth was enhanced also without electrical stimulation [137]. Moreover, these findings were also confirmed by Alon et al. [138] and Nissan et al. [139]. In parallel, substrates coated with Ag NPs demonstrated to act as scaffolds to sustain the growth of neuroblastoma cells. This seems related to the surface properties of Ag NPs, that present anchoring sites for neuroblastoma cells and significantly increase the neurite outgrowth. Furthermore, this result was conceivably attributed to particle density with a maximum effect for the 45 nanoparticles/µm^2^ sample, suggesting that the coating with Ag NPs combined with an adequate topography could be attractive for neuronal repair [140].

## 6. Antimicrobial Properties

Metallic nanoparticles and metal oxides, such as Ag, Au, copper (Cu), titanium (Ti), and zinc (Zn), are well known to inhibit the growth of several species of bacteria, fungi, and viruses. Moreover, nanoparticles shapes and sizes have been described as key elements for the control of bacterial growth. Furthermore, microorganisms resistant to the most used antibiotics, that represent a great threat to human health, are generally sensitive to these aspecific components. For these reasons, the combination of proliferation enhancement and antimicrobial properties renders metallic and metal oxides nanoparticles very important for biomedical applications to prevent microbial infections and to promote tissue regeneration [118].

In particular, the nanoparticle properties could affect their antimicrobial potency. Smaller sized nanoparticles demonstrated to cause higher bacterial inhibition [141]. This could be attributed to the size of bacterial cell, which is in micrometer range, while their membrane pores are in nanometric dimensions and could be more easily entered by smaller nanoparticles to denature the intercellular proteins and consequently kill the bacteria [118].

Krishnaraj et al. have reported the anti-microbial activity against *E. coli* and *Vibrio cholera* of 20–30 nm Ag NPs [142]. Similarly, hydrophobic and cation-functionalized Au NPs showed a strong bactericidal activity against both Gram-negative and Gram-positive multiple drug resistant bacteria [143].

Cu nanoparticles (Cu NPs) also possess a great prospective to act as anti-microbial agents. The anti-microbial activity of Cu NPs was evaluated against bacteria such as *Micrococcus luteus*, *S. aureus*, *E. coli*, *K. pneumoniae*, and *P. aeruginosa*, and against fungi such as *Aspergillus flavus*, *Aspergillus niger*, and *Candida albicans* [144]. Table 2 summarizes the antimicrobial activities and studies caried out on various types of nanoparticles containing inorganic compounds.

The antimicrobial activity mechanisms are not yet clearly understood. It is hypothesized that nanoparticles accumulate near the microbial cell membrane and enter thanks to membrane damage or cavity formation on the membrane. After entering the bacterial cell membrane, nanoparticles produce free radicals or interact with proteins inside the bacterial cells, thus determining enzyme inactivation and, consequently, cell death (Figure 9) [118].

## 7. Conclusions and Future Perspectives

Extensive studies were devoted to the development of scaffolds doped with inorganic components. They have been proposed as medical devices in regenerative medicine and tissue engineering for both soft and hard tissues applications.

The inorganic nanomaterials possess unique physical, chemical, optical, mechanical, and electrical properties, which render them interesting substrates especially in biomedical field. Inorganic nanoparticles actually play a crucial role in disease diagnostics, therapy, tissue engineering and theranostics. The size, shape, morphology, and surface chemistry of the nanoparticles have also demonstrated to alter their properties and behavior in biological systems.

Among different types of nanomaterials, inorganic nanoparticles, and in particular clays, carbon-based nanomaterials, metal oxides and metallic nanoparticles, allowed a significant improvement in the scaffold properties, together with direct effect on the growth of different types of cells and antimicrobial effectiveness. Therefore, as it has been shown in this review, inorganic nanomaterials can be successfully applied for skin, orthopaedic, and neural tissues regeneration. Despite this, the assessment of nanotoxicological profile still remains an open question. This is particularly challenging for non-degradable materials, such as clays and carbon-based nanomaterials, which potentially could accumulate in the tissues causing long term effects. These effects should be deeply studied in large animals to better define the immune response. In fact, currently, especially for what concern carbon-based nanomaterials, the majority of the investigations has been performed on mice models, which are characterized by tremendous differences in the immunological system compared to humans. Moreover, all the materials obtained by chemical synthesis (metal oxides, metallic nanoparticles, carbon-based nanomaterials and synthetic clays) could suffer the presence of impurities that significantly alter the reliability of the obtained results. As final remark, clay-based and carbon-based nanomaterials are also used as drug delivery systems due to their hollow structure. Moreover, for all the nanomaterials here described there is also the possibility to be functionalized bearing active ingredients that act synergically to potentiate reparative and antimicrobial effectiveness.

In conclusion, big steps have been made for employing different inorganics for the production of scaffolds in the field of tissue engineering. Additionally, many studies highlighted the potentiality of these systems in controlling the specific tissue functionality. However, preclinical long-term studies focused on prolonged treatment with inorganics combined with scaffolds are still required. Therefore, these factors need to be investigated in the future, opening an opportunity for further clinical trials.

## Figures and Tables

**Figure 1 pharmaceutics-14-01127-f001:**
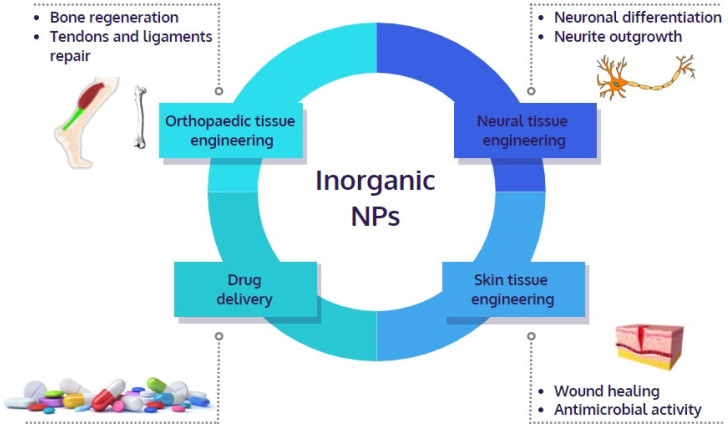
Use of inorganic nanoparticles (NPs) in tissue engineering.

**Figure 2 pharmaceutics-14-01127-f002:**
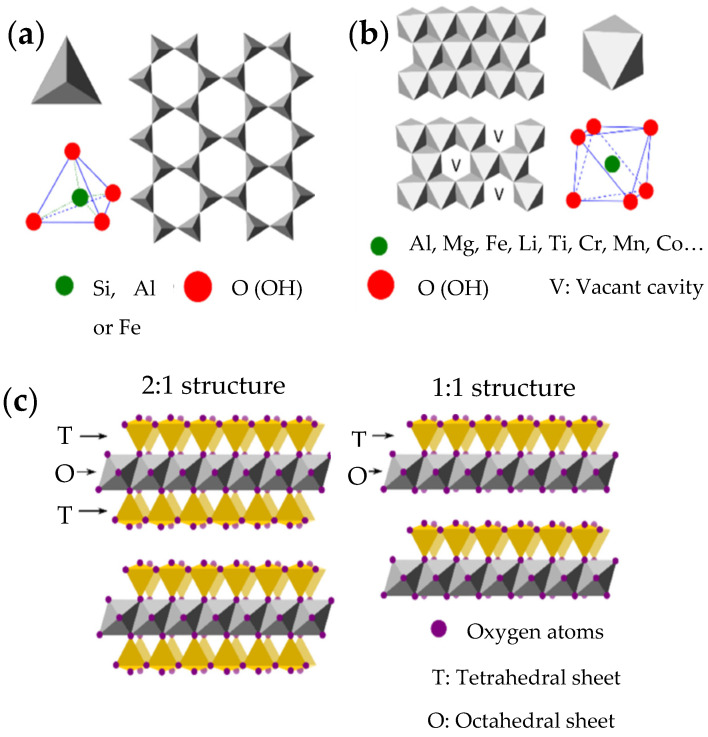
Schematic representation of (**a**) basic phyllosilicate tetrahedron and spatial disposition of tetrahedral sheets (T); (**b**) basic phyllosilicate octahedron and spatial disposition of octahedral sheets (O); (**c**) phyllosilicates layer structures 2:1 (T:O:T) and 1:1 (T:O). Adapted with permission from [13]. CC BY 4.0.

**Figure 3 pharmaceutics-14-01127-f003:**
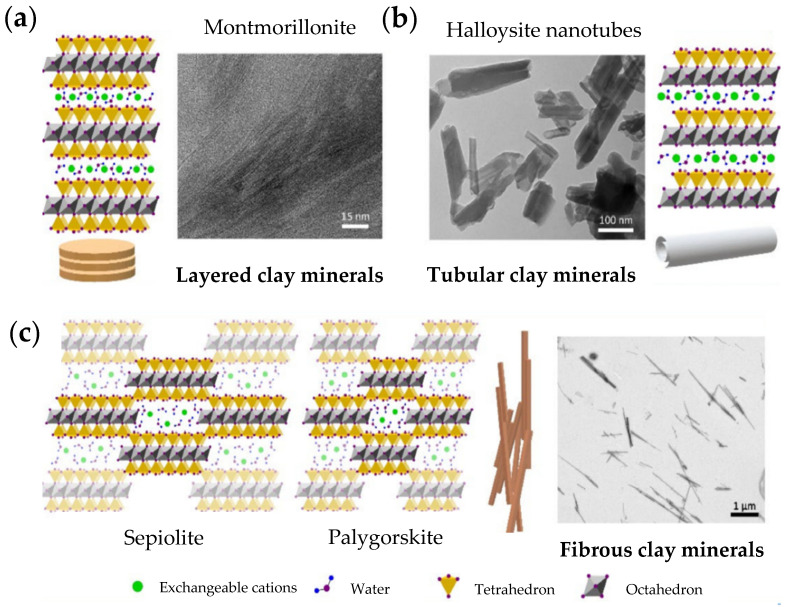
Structure and microphotographs of (**a**) layered clay minerals, (**b**) tubular clay minerals, and (**c**) fibrous clay minerals. Adapted with permission from [13]. CC BY 4.0.

**Figure 4 pharmaceutics-14-01127-f004:**
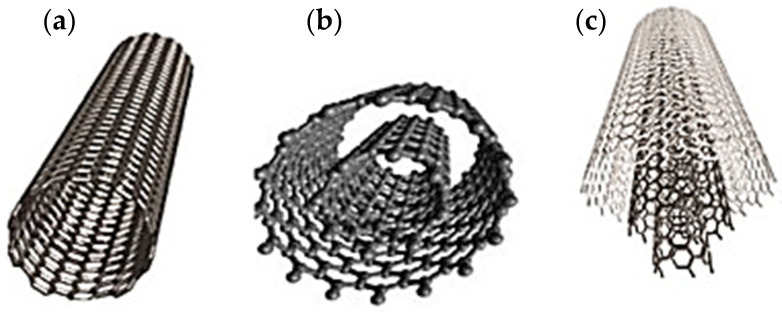
Structure of single-walled (**a**), double-walled (**b**), and multi-walled (**c**) CNTs. Adapted with permission from [37]. Copyright 2020 Elsevier B.V.

**Figure 5 pharmaceutics-14-01127-f005:**
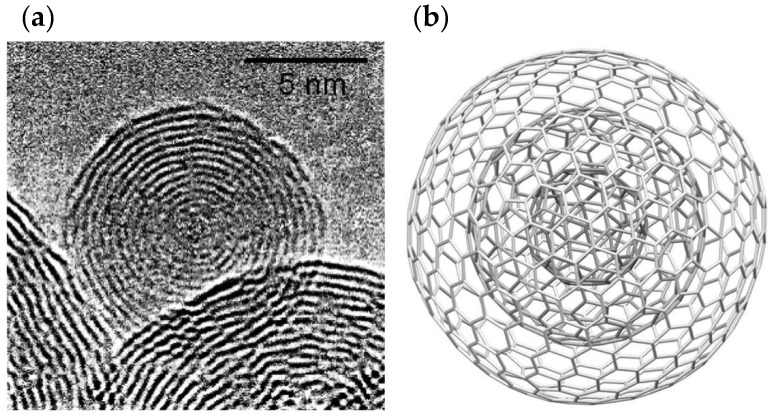
Structure of CNOs seen with (**a**) High-Resolution Transmission Electron Microscopy image, and (**b**) 3D graphical image. Adapted with permission from [39]. Copyright 2017 Elsevier B.V.

**Figure 6 pharmaceutics-14-01127-f006:**
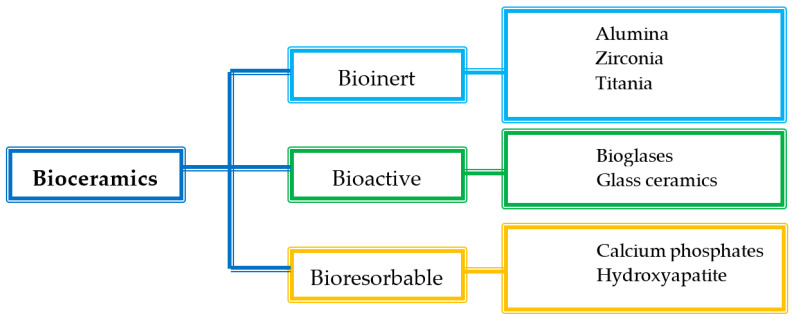
Classification of bioceramics.

**Figure 7 pharmaceutics-14-01127-f007:**
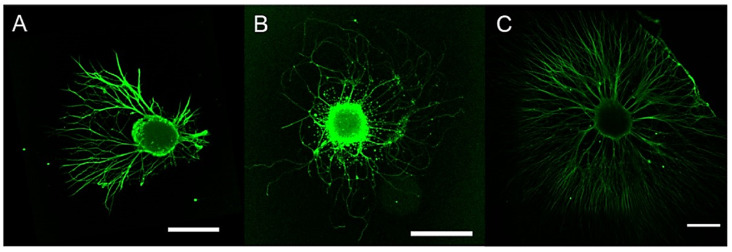
CLSM images (scale bar = 500 µm) showing the growth of extending neurites from dorsal root ganglia (DRG) directed by nerve growth factor (NGF) gradients generated by poly-L-lactic acid/iron oxide NPs (cNPs). DRG were subjected to (**A**) NGF combined with cNPs, (**B**) cNPs alone, and (**C**) NGF containing medium. (**A**) The combination of NGF and cNPs directed the extension of neurites from DRG. (**B**) Control cNPs had no effect on the directional outgrowth of neurites, and (**C**) NGF containing medium did not impart a growth preference on neurites. Adapted with permission from [89]. Copyright 2022 American Chemical Society.

**Figure 8 pharmaceutics-14-01127-f008:**
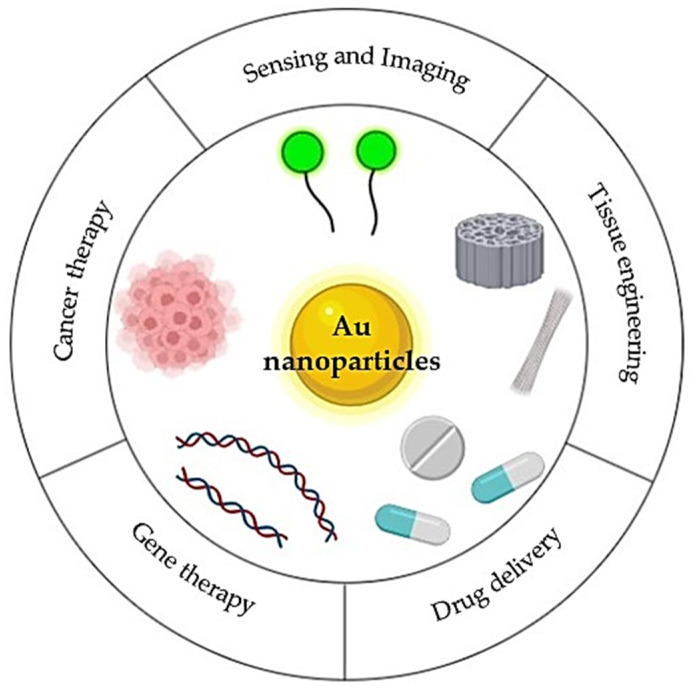
Applications of Au NPs in the biomedical field. Adapted with permission from [112]. Copyright 2022 John Wiley & Sons, Inc.

**Figure 9 pharmaceutics-14-01127-f009:**
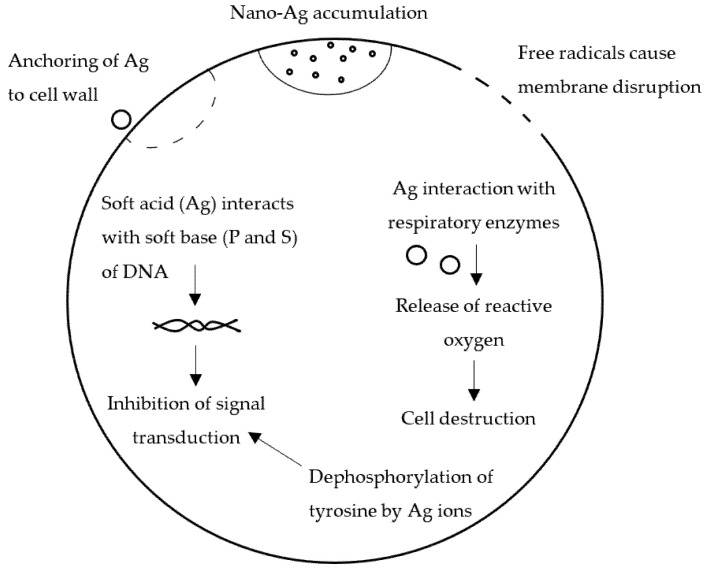
Mechanisms of the Ag NPs antimicrobial effect. Adapted with permission from [118]. CC BY 2.0.

**Figure 10 pharmaceutics-14-01127-f010:**
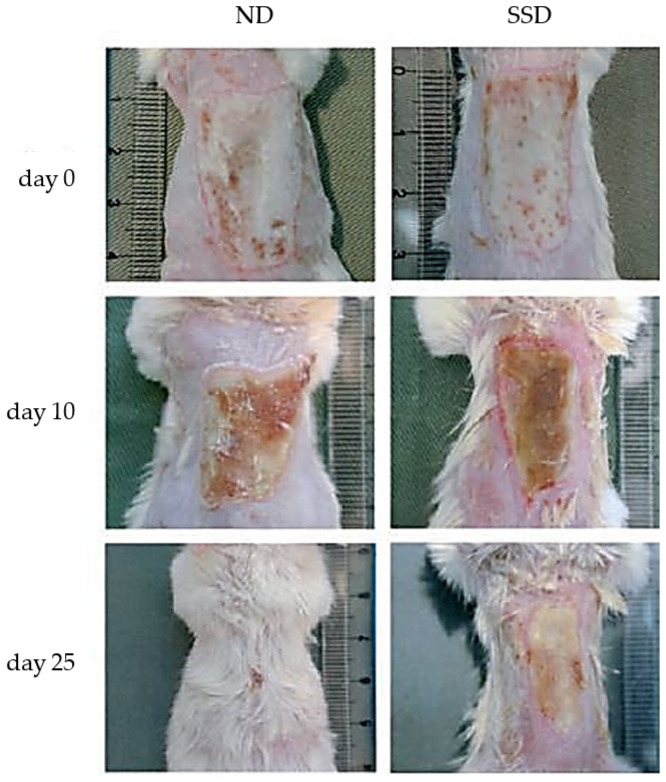
Images of burn wound from mice treated with Ag NPs (ND) and silver sulfadiazine (SSD) at different times of wound healing. Adapted with permission from [130]. Copyright 2022 John Wiley & Sons, Inc.

**Table 1 pharmaceutics-14-01127-t001:** Mechanical properties of pure polymeric materials scaffolds in respect to the ones loaded with carbon-based nanomaterials.

Material	Young’s Modulus (GPa)	Tensile Strength (MPa)	References
UHMWPE ^1^	1.51 ± 0.01	18.62 ± 0.16	[43]
f-SWCNTs/UHMWPE ^2^ (0.01 wt%)	1.69 ± 0.01	19.20 ± 0.11	
f-SWCNTs/UHMWPE (0.1 wt%)	1.74 ± 0.006	28.00 ± 0.11	
GelMA ^3^	7.12 ± 3.1	157.81 ± 3.91	[44]
CNOs/GelMA	41.19 ± 3.78	356.12 ± 3.67	
AN-PEEK ^4^	9.9 ± 0.3	195.4 ± 1.1	[45]
AN-PEEK/CNOs (1.5 wt%)	24.1 ± 0.8	351.1 ± 3.5	
AN-PEEK/CNOs (2.5 wt%)	31.3 ± 0.4	552.9 ± 6.1	
AN-PEEK/CNOs (5 wt%)	43.2 ± 1.1	891.4 ± 8.2	
PCL	0.041 ± 0.0009	41.20 ± 0.89	[46]
PCL/CNOs (0.2 wt%)	0.06 ± 0.0013	60.10 ± 1.27	
PCL/CNOs (0.5 wt%)	0.084 ± 0.0011	84.70 ± 1.14	

^1^ UHMWPE: ultra-high molecular weight polyethylene; ^2^ f-SWCNTs: functionalized single-walled CNTs; ^3^ GelMA: gelatin methacryloyl; ^4^ AN-PEEK: anilinated-poly (ether ether ketone).

**Table 2 pharmaceutics-14-01127-t002:** Antimicrobial activity of nanoparticles based on inorganic compounds.

Nanoparticles	Antimicrobial Activity	References
Ag	*Salmonella typhi, Salmonella paratyphi*, *V. cholera* and *S. aureus*	[145]
Ag	*B. subtilis, Klebsiella planticola*, *K. pneumonia, Serratia nematodiphila*, and *E. coli*	[146]
Ag	*B. subtilis, K. pneumonia, E. coli*, *P. aeruginosa* and *S. aureus*	[147]
Au	BCG ^1^ and *E. coli*	[148]
Au	*S. aureus, E. coli, K. Pneumonia* and *P. aeruginosa*	[149]
Cu	*E. coli* and *C. albicans*	[144]
Cu	*E. coli, K. pneumonia* and *S. aureus*	[150]
Cu	*S. typhi, B. subtilus, S. aureus, K. pneumoniae* and *E. coli*	[151]
Cu	*S. aureus* and *P. aeruginosa*	[152]
Zn	*E. coli* and *S. aureus*	[153]
TiO_2_	*E. coli* and *S. aureus*	[154]
TiO_2_	*E. coli, S. aureus* and *K. pneumonia*	[155]
Fe_2_O_3_	*S. aureus, E. coli, P. aeruginosa* and *Serratia marcescens*	[155]
Iron	*E. coli, Salmonella enterica, Proteus mirabilis* and *S. aureus*	[156,157]

^1^ Bacillus Calmette-Guérin.

## Data Availability

Data sharing is not applicable to this article.

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
