# Peer review of "Inorganic Nanomaterials in Tissue Engineering"

_pharmaceutics, 2022, doi:10.3390/pharmaceutics14061127_

Round 1

Reviewer 1 Report

The authors have prepared a review article and discussed the synthesis, characterizations, and tissue engineering applications of a variety of inorganic fillers (clays, metal oxides, and metallic nanoparticles) reinforced scaffolds. Initially, the review shows a brief introduction to inorganic nanofillers, tissue repair, and regeneration, followed by biodegradable biomaterials and their scaffolds for tissue engineering applications. A significant effort was made to prepare the review, which is quite interesting and unique. This article is impressive for the reviewer and audience of the nanotechnology community as well as material science. This review would show a significant impact on the tissue engineering and materials science community. The reviewer recommends this work be published in Pharmaceutics after a major revision.

The reviewer has the following comments  

  1. Rephrase the following sentence from the abstract

For this reason, the field of tissue engineering is of great interest as it allows the development of scaffolds mimicking the tissues extracellular matrix.

  1. Insert some infographics to make the manuscript interesting to the reviewers and readers. Also, able to secure future citations and visibility.
  2. It is well known that carbonaceous materials (carbon nanotubes, nano-onions, etc.) are outstanding nanofillers for orthopedic and tissue engineering and drug delivery. So, it is suggested to include carbonaceous materials in the revised manuscript
  3. The introduction section is very short and should be improved entirely so that the reader can clearly identify the scientific problems solved by this research. The introduction of nanofillers specifically, carbonaceous materials reinforced nanocomposites must be reported. For instance, cite the following research work,

https://doi.org/10.1016/j.colsurfb.2021.111819,

  1. Make a figure/table of mechanical properties for various nanofillers-based biomaterials. Because mechanical properties of biomaterials are major characteristics for biomedical applications. The authors should use the following very important articles

https://doi.org/10.1016/j.jmbbm.2021.104554, https://doi.org/10.3390/ph14040291, https://doi.org/10.3390/pharmaceutics12121208, DOI: 10.1557/jmr.2020.23, https://doi.org/10.1039/D1EN00354B,

It would be more realistic to cover such kind of research work in the current manuscript. Which will enrich the quality of the current manuscript as well as inquisitiveness to the readers.

  1. A copyright statement should be added to the caption of all the figures if you have taken the figures from the literature.
  2. The authors should describe and include the "Future prospective” section in the revised manuscript.

Author Response

REVIEWER #1

The authors with to thank the reviewer for the comments aimed at improving the quality of the paper.

  1. Rephrase the following sentence from the abstract: For this reason, the field of tissue engineering is of great interest as it allows the development of scaffolds mimicking the tissues extracellular matrix.

A: The sentence has been rephrased.

  1. Insert some infographics to make the manuscript interesting to the reviewers and readers. Also, able to secure future citations and visibility.

A: An infographic on the use of inorganic nanoparticles in tissue engineering has been added.

  1. It is well known that carbonaceous materials (carbon nanotubes, nano-onions, etc.) are outstanding nanofillers for orthopedic and tissue engineering and drug delivery. So, it is suggested to include carbonaceous materials in the revised manuscript

A: We included in the draft a new chapter about carbonaceous materials, in particular carbon nanotubes and carbon nano-onions, and their applications in skin, orthopaedic and neural tissue engineering.

  1. The introduction section is very short and should be improved entirely so that the reader can clearly identify the scientific problems solved by this research. The introduction of nanofillers specifically, carbonaceous materials reinforced nanocomposites must be reported. For instance, cite the following research work: https://doi.org/10.1016/j.colsurfb.2021.111819

A: The introduction has been improved and the suggested research work has been added to the draft.

  1. Make a figure/table of mechanical properties for various nanofillers-based biomaterials. Because mechanical properties of biomaterials are major characteristics for biomedical applications. The authors should use the following very important articles

https://doi.org/10.1016/j.jmbbm.2021.104554,

https://doi.org/10.3390/ph14040291,

https://doi.org/10.3390/pharmaceutics12121208,

DOI: 10.1557/jmr.2020.23,

 https://doi.org/10.1039/D1EN00354B.

It would be more realistic to cover such kind of research work in the current manuscript. Which will enrich the quality of the current manuscript as well as inquisitiveness to the readers.

A table containing the mechanical properties of the polymeric scaffolds enriched with carbon based nanomaterials has been added, also with the suggested literature

  1. A copyright statement should be added to the caption of all the figures if you have taken the figures from the literature.

A copyright statement with the corresponding permission code has been added to the caption of all figures taken from literature. The only exceptions are figure 1 and figure 6, which we created.

  1. The authors should describe and include the "Future prospective” section in the revised manuscript.

Accordingly to the reviewer’s comments, the conclusions paragraph has been integrated with future perspective and entitled “conclusions and future perspective”.

Reviewer 2 Report

The presented review covers a small part of the possibilities of using
nanomaterials in tissue engineering. The review is structured logically,
there are illustrations, although I would recommend the authors to
include a general scheme for the use of nanomaterials in tissue
engineering. Some recent articles are missing, although they reflect the
development of this topic. For example on the use of clay nanomaterials: ACS Biomater. sci. Eng.
2019, 5, 8, 4037–4047;  Carbohydrate Polymers, V
269, 2021, 118311, DOI 10.1016/j.carbpol.2021.118311.
And also on the use of iron oxide nanoparticles: Journal of Nanomaterials 2021(7):1-14
DOI:10.1155/2021/3762490

Author Response

REVIEWER #2

The authors with to thank the reviewer for the comments aimed at improving the quality of the paper.

- I would recommend the authors to include a general scheme for the use of nanomaterials in tissue
engineering.

A: A general scheme for the use of inorganic nanoparticles has been added.

- Some recent articles are missing, although they reflect the development of this topic. For example on the use of clay nanomaterials: ACS Biomater. sci. Eng. 2019, 5, 8, 4037–4047; Carbohydrate Polymers, V 269, 2021, 118311, DOI 10.1016/j.carbpol.2021.118311.

A: The suggested articles have been added to the draft.

- And also on the use of iron oxide nanoparticles: Journal of Nanomaterials 2021(7):1-14
DOI:10.1155/2021/3762490

A: The suggested article has been added to the draft

Reviewer 3 Report

This manuscript is aimed at overview of the Inorganic nanomaterials including clay, metal oxides and metallic nanoparticles, and their application in tissue engineering. The work summarises most of recent progress in the field , which is valuable for researchers. Reviewer recommends publication of this paper in current formate. In general, pharmaceutics indicate drug delivery, pharmacokinetics, biopharmaceutics, pharmacogenetics, therefore, based on the content, I would suggest the manuscript is more suitable for publication in Nanomaterials rather than in Pharmaceuticals. 

Author Response

REVIEWER #3

This manuscript is aimed at overview of the Inorganic nanomaterials including clay, metal oxides and metallic nanoparticles, and their application in tissue engineering. The work summarises most of recent progress in the field , which is valuable for researchers. Reviewer recommends publication of this paper in current formate. In general, pharmaceutics indicate drug delivery, pharmacokinetics, biopharmaceutics, pharmacogenetics, therefore, based on the content, I would suggest the manuscript is more suitable for publication in Nanomaterials rather than in Pharmaceuticals. 

The manuscript has been revised including a paragraph focused on the carbon based nanomaterials.

The conclusions have been implemented with reference to the future perspective and info about the possibility to load these nanostructured materials with drugs has been added.

Round 2

Reviewer 1 Report

The authors have clarified all the concerns, and the quality of the revised manuscript was ameliorated. Thus, I recommend the acceptance of the revised manuscript in its present form. 

Congratulations!

Reviewer 2 Report

All recommendations were taken into account. The article can be accepted in its current form for publication.

Reviewer 3 Report

The authors have amended the content based on my questions that I pointed out in previous review report. 

Round 3

Reviewer 1 Report

The revised manuscript can be acceptable in its current form.